# Fine Structure of the Visual System of *Arge similis* (Hymenoptera, Argidae)

**DOI:** 10.3390/insects13020152

**Published:** 2022-01-30

**Authors:** Chao Wen, Zijian Pan, Shiping Liang, Liming Shen, Xiujun Wen, Cai Wang

**Affiliations:** Guangdong Key Laboratory for Innovation Development and Utilization of Forest Plant Germplasm, College of Forestry and Landscape Architecture, South China Agricultural University, Guangzhou 510642, China; wenchaoxr@stu.scau.edu.cn (C.W.); zijain-pan@stu.scau.edu.cn (Z.P.); lsp54542727@163.com (S.L.); sliming5893@163.com (L.S.)

**Keywords:** ommatidia, photoreceptors, retina, sawfly, vision

## Abstract

**Simple Summary:**

*Arge similis* is a notorious pest on rhododendron plants. The fine structure of the compound eyes of *A*. *similis* adults were examined by light, scanning, and transmission electron microscopy. Each ommatidium consists of a lamellated cornea, a crystalline cone with eucone type, and a centrally fused rhabdom composed of eight retinular cells.

**Abstract:**

External morphology and ultrastructure of the visual system of *Arge similis* (Vollenhoven, 1860) adults were investigated by light microscopy, scanning electron microscopy, and transmission electron microscopy. Each compound eye contains 2022 ± 89 (mean ± SE) facets in males and 2223 ± 52 facets in females. *Arge similis* has an apposition kind of compound eye composed of a cornea, a crystalline cone of four cone cells, and a centrally fused rhabdom made up of the rhabdomeres of eight large retinular cells. Each crystalline cone is surrounded by primary and secondary pigment cells with black spherical screening pigment granules measuring 0.60 ± 0.02 and 0.41 ± 0.01 μm in diameter, respectively. Based on our findings, the compound eye of *A. similis* can be expected to exhibit high adaptability to light intensity changes.

## 1. Introduction

As the main visual organ of insects, the compound eye has a very long history that may stretch back to the neo-proterozoic [1,2]. Insects use their eyes to sense the size, shape, and color of an object, and therefore their eyes play essential roles in feeding, foraging, oviposition, homing, and enemy avoidance [3,4]. 

A compound eye’s structure reflects its visual capacity [5,6] and is attuned to the lifestyle of an insect [7]. More than a hundred years ago, Exner [8] divided the compound eyes into superposition and apposition eye types based on their optical and structural differences. In superposition eyes, the dioptric apparatus (cornea and cones) is separated from the retinular cells by a pigment-free space in the ommatidium known as the “clear-zone” [9]. In this type of eye, a single ommatidial rhabdom of the retina can receive light from hundreds of corneal lenses, and insects that possess superposition eyes are sensitive to low-intensity light. By contrast, the apposition eye lacks the “clear-zone”, and the rhabdom of one ommatidium can only receive light from a single corneal lens directly above it. Most diurnally active insects possess the apposition type of eye [10,11]. However, some nocturnal dipterans, such as *Arachnocampa luminosa* [12], *Clogmia albipunctata* [13], some mosquitoes [14], and the nocturnal hymenopteran *Megalopta genalis* [15], have apposition eyes that also function under dim light.

*Rhododendron* L. is a genus of various economically important shrubs with high ornamental value. It is one of the largest genera of flowering plants comprising ca. 1000 species [16,17], which are widely distributed in China, Japan, and many other regions of the world [18,19]. *Arge similis* (Vollenhoven, 1860), also known as rhododendron sawfly, is a notorious pest of rhododendron plants with its larvae feeding on the leaves of *Rhododendron* species such as *Rhododendron pulchrum*, *R. indicum*, and *R. mucronatum* [20]. This pest poses a serious threat to the development and ornamental values of rhododendron plants [20]. *Arge similis* is a diurnal insect (i.e., the adult is active during the daytime) that feeds on rhododendron leaves to satisfy nutritional requirements for reproduction [17,21,22].

The ecology, pest status, and biological characteristics of *A. similis* have been well studied [17,20]. However, the structure of the photoreceptors of *A. similis* is still unclear. As far as we know, there is only one study on the structure and function of the larval eye of a leaf wasp [23]. Herein, we explore the external morphology and microanatomical structure of the eyes of *A. similis* by light microscopy (LM), scanning electron microscopy (SEM), and transmission electron microscopy (TEM).

## 2. Materials and Methods

### 2.1. Insects

Third instar larvae of *A*. *similis* were collected in the Wutong Mountain Scenic Spot (113°17’ E, 22°23’ N), Shenzhen, Guangdong, China. The collected insects were reared on leaves of *Rhododendron simsii* grown in the College of Forestry and Landscape Architecture, South China Agricultural University. Host plants and insects were kept under relatively constant conditions (temperature: 25 ± 2 °C; relative humidity: 75 ± 10%; photoperiod: 14:10 h [L:D]). Pupae were collected daily and transferred into a plastic container (bottom diameter: 101 mm; height: 89 mm) filled with 3 cm thick moist soil. The emerging insects were fed with a 20% honey-water solution each day. Adults of 3–5 days of age were collected and used for the analyses.

### 2.2. Compound Eye Preparation for Scanning Electron Microscopy (SEM)

For the SEM analysis, three females and three males of *A. similis* were used. The gender of *A. similis* was differentiated using the method provided by Shi [21]. In brief, the end of the abdomen of female adults has a small gap, whereas the gap is absent for male adults [21]. After the gender of the adults was determined, the heads of alive insects were severed during daytime hours under room light using fine scissors and immediately fixed in 2.5% glutaraldehyde for 24 h. The heads were then dehydrated in a graded series of 70%, 80%, 90%, 95%, and 100% ethanol with stays of 10 min in each concentration. After drying in an oven, the heads were stuck on an aluminum stub and sputter-coated with gold to a thickness of 0.02 μm at 30 mA current for 100 s using a Leica EM ACE600 (Leica, Wetzlar, Germany). High-resolution micrographs of the surfaces of compound eyes were taken using an SEM (EVO MA15, Carl Zeiss AG, Oberkochen, Germany).

### 2.3. Compound Eye Preparation for Transmission Electron Microscope (TEM)

For the TEM analysis, *A. similis* were placed in an incubator (PQX-450B-22H, Ningbo Laifu Technology Co., Ltd, Ningbo, China) for ~3 h in the darkness to render them dark-adapted. Controls were kept under normal light conditions before fixing. All insects were decapitated during the daytime. The eyes were fixed in 4% glutaraldehyde in PBS buffer (0.1 M, pH 7.4), undergoing three changes (15 min each time) and post-fixed in 1% O_S_O_4_ for approximately 24 h. The samples were then kept in 0.1 M cacodylate buffer for 1 h at room temperature, and dehydrated in a graded series of ethanol (15 min for each concentration) and transferred into absolute acetone for 20 min. After that, the specimens were placed in a 3:1 mixture of absolute acetone and Spurr’s resin overnight, and transferred to 1:1 and 1:3 mixtures of absolute acetone and pure Spurr’s resin (12 h for each mixture). The samples were hardened at a temperature of 70 °C for 24 h in an oven. Frontal, longitudinal, and tangential serial sections (70–90 nm) were cut using a diamond knife on an ultramicrotome (Leica UCT, Leica, Wetzlar, Germany). Ultrathin sections were picked up on uncoated 100 mesh or formvar-coated copper slot grids and stained with 2% aqueous uranyl acetate for 15 min. Observations took place under a transmission electron microscope (Talos L120C and Talos F200S, Talos, MA, USA) at 80 kV.

### 2.4. Compound Eye Preparation for Light Microscopy (LM)

Semithin sections (300 nm) were cut on an ultramicrotome with glass knives. The sections were stained with an aqueous solution of toluidine blue (0.5%) on a hot plate for 100 s, and observed under an inverted fluorescent microscope (Leica, DMI8, Leica, Wetzlar, Germany).

### 2.5. Data Obtaining and Analysis

All histological measurements based on high-resolution photographs were measured using Image J version 1.52 software (Rasband, W.S., U.S. National Institutes of Health, Bethesda, Rockville, MD, USA). Total numbers of ommatidia per eye, dimensions of compound eyes and facets, and the diameters of the ocelli were determined using SEM. In addition, TEM was used to measure the dimensions of pigment granules and cone cells, count the number of cone cells and retinal cells, and ascertain the presence or absence of certain organelles like mitochondria and vesicles. 

## 3. Result

### 3.1. External Morphology of the Compound Eye and Ocelli in Arge similis

*Arge similis* is of blue body coloration with a body length of 10 mm (Figure 1A). They possess an oval-shaped compound eye on each side of the head, slightly raised above the surrounding cuticle (Figure 1B). The area of the compound eyes is 0.59 ± 0.01 mm^2^ in males and 0.60 ± 0.02 mm^2^ in females (Table 1). Respective major (dorso-ventral direction) and minor axes (anterior-posterior direction) of the eyes of the males are 1165.67 ± 24.70 μm and 607.11 ± 27.43 μm, while the corresponding ones of the female individuals were 1192.20 ± 8.73 μm and 603.66 ± 3.80 μm (Table 1). 

Each eye contains 2022 ± 89 facets in males and 2223 ± 52 facets in females (Table 1). The average facet diameter in males is 17.40 ± 0.19 μm, while the average facet diameter of females is 18.26 ± 0.17 μm (Table 1). The vast majority of the facets are hexagonal, but a small number of facets located near the dorsal edge of the eye are of pentagonal shapes (Figure 1E,F). The outer surfaces of ommatidium were smooth and devoid of corneal nipples or interfacetal hairs (Figure 1E,F). Three dorsal ocelli are located on the dorsal central region of the head between the compound eyes (Figure 1D). The three ocelli were arranged in a regular triangle (at an angle of approximately 60° between them), and the size of the median ocellus (diameter = 106.92 μm) is larger than the outer/lateral ocelli (diameter = 87.14 μm).

### 3.2. Ultrastructure of the Compound Eyes in Arge similis

Compound eyes of *A. similis* are composed of two distinct regions: the dioptric apparatus with cornea and cones and the photoreceptive layer with retinula and supporting cells (Figure 2).

### 3.3. Dioptric Apparatus

The outermost layer of the ommatidium is the lamellated cornea (Figure 3A). The corneal lens is composed of a layered structure that results from different orientations of successive chitin micelles (as shown for insect cuticle generally) by Bouligand [24] (Figure 3B). The concave apex of the crystalline cone connects the inner convex surface of the corneal lens (Figure 3A). Since there is no “clear-zone” present in the ommatidium of *A. similis* (Figure 2 and Figure 3A), it conforms to the apposition eye type. The crystalline cone with a length of 22.10 ± 0.76 μm is below the corneal lens (Table 1, Figure 3A). As in other hymenopterans, the crystalline cone is composed of four cone cells (so-called Semper cells), and each cone cell contributes, like the segments of an orange, one-quarter to the crystalline cone (Figure 3D). The cone cells formed four cone cell roots (Figure 4). The mottled nuclei of the cone cells are located at the distalmost part of the crystalline cone. The crystalline cone is of the eucone type, and each cone cell has an electron-empty peripheral cytoplasmic region, in which organelles are sparse (Figure 3B). The crystalline cone is surrounded by primary pigment cells and secondary pigment cells, which isolate each crystalline cone from its neighbors (Figure 3C,D).

### 3.4. Photoreceptive Layer

The photoreceptive layer consists of retinular cells and their rhabdoms plus accompanying screening pigment cells that shield each ommatidial group of retinular cells from the neighboring ommatidia (Figure 5A,B and Figure 6). An extensive palisade is developed toward the periphery of the retinular cells (Figure 5C,D). A total of eight retinular cells (R1–R8) contribute their rhabdomeres to form the centrally fused rhabdom (Figure 4). Each retinular cell has a rhabdomere made up of finger-like microvilli on its inner central side to join neighboring rhabdomeres, thereby creating the centrally fused column of the rhabdom (Figure 4). The rhabdom palisade is made up of irregularly shaped cisternae in the peripheral cytoplasm of the retinular cells (Figure 5). The distal pigment granules were far from the rhabdom under normal light intensity (Figure 5C), while the screening pigment granules of retinular cells were near the rhabdom during dark adaptation (Figure 5D). Typical cellular organelles of the retinular cells identified in the retinular cells’ cytoplasm were mitochondria and extensive endoplasmic reticulum (Figure 5C).

### 3.5. Pigments

Consistently, two primary pigment cells (PPCs) enveloped each crystalline cone and their four cone cells from their distal position close to the cornea (Figure 3A). The nuclei of the PPCs were elongated and positioned close to the cornea. In addition, secondary pigment cells (SPCs) were found to extend into the photoreceptive layer of ommatidium (Figure 3B,D). The electron densities or chemical compositions of the pigment granules located in the PPCs and SPCs appear different from each other (judging by their coloration). A distinction between pigment granules and what seemed to be lipid-like inclusions was not always easy to make for the PPCs (Figure 3B). The pigment granules of the PPCs (diameter: 0.60 ± 0.02 μm) are much larger than those of SPCs (diameter: 0.41 ± 0.01 μm). The retinular cells contain their own screening pigment granules of about 0.34 ± 0.02 μm in diameter. It is worth noting the elongated rodlike shapes (rather than the more usual spherical shapes) of the pigment granules, which may be the result of poor fixation or errors during the process of measuring the diameters of the pigment particles. They also may represent aberrant, malformed, or even a subgroup of screening pigment grains.

## 4. Discussion

Previous studies showed that the eyes in nocturnal Hymenoptera are larger than those of diurnal Hymenoptera [25]. For example, the compound eyes of the nocturnal halictid bee *Megalopta genalis* consist of about 4880 facets in males, with facet diameter ranging from 28–36 μm [15], which is significantly greater than the corresponding values in the diurnal *A. similis*. However, the size of the compound eyes in *A. similis* is about 15 times that of the tiny diurnal wasp *Trichogramma evanescens*, whose long axes in dorsoventral direction are 63.39 and 71.11 μm in males and females, respectively [26]. The sizes of compound eyes were also positively correlated with body size in Lepidoptera, bees, ants, and beetles, and not only species with nocturnal or diurnal lifestyles [7,11,25,27,28]. There is no doubt that a larger compound eye area receives more light under identical illumination and may be more sensitive to dim light [10,29,30]. Also, other aspects, such as the number of microvilli in a rhabdom and photopigment density in photoreceptors, length and width of the rhabdom, membrane turnover processes, and energy requirements, may be involved in the performance of the photoreceptors [31,32,33]. Interestingly, the number of ommatidia and the area of compound eyes of females are larger than that of male *A. similis*. Large compound eyes indicate that the insects can be expected to exhibit a high-quality vision ability [30,34,35,36], and may help gravid females to perceive suitable host plants to satisfy nutritional requirements and oviposition. In addition, some Lepidoptera insects search host plants through a combination of visual and olfactory stimuli [37]. A previous study showed that *A. similis* females had obvious electrophysiological responses to four compounds of volatiles of *R. pulchrum*, indicating that the olfactory organ may also play an important role in finding a host for *A. similis* females [17].

Compound eyes of arthropods occur in two major types: apposition eyes and superposition eyes [8,38]. Unsurprisingly, the compound eye of *A. similis* is of the apposition type. This eye type is generally diagnostic for Hymenoptera and has been reported in *Apis mellifera* [39], *Megalopta genalis* [15], *Cataglyphis bicolor* [40], and the wasp *Sphex cognatus* [41]. Behavioral studies also showed that *A. similis* is an insect with a typically diurnal lifestyle because its feeding, mating, and spawning occur during the photophase [17].

The laminated cornea lens suggested that it was deposited by growth layers [24] and contributes little to the optical limitations of the cornea. The shapes and dimensions of the corneal lenses are associated with the amount of light, which can enter the underlying ommatidium to be available for the photoreceptive microvillar membranes of rhabdom [7,42,43]. According to a study by Barlow [44], the theoretical limit of the diameter of a corneal lens to be effective in apposition eyes should be about 8–10 μm. The diameter of the facet is 18.26 ± 0.17 μm for *A. similis* females and 17.40 ± 0.19 μm for males, which are much higher than the values predicted by Barlow [44]. The cornea of *A. similis* is larger and relatively thick, which could be an adaptation to provide protection for the compound eye against overexposure to high light intensity, physical damage [45], and harmful radiation [46]. Similarly, the eyes of the sun-loving beetles *Curis caloptera* have a thick cornea, allowing them to be active during the day under sunlight exposure [47].

In the retinular cells of *A. similis*, extensive palisade was found around the rhabdom, and this palisade may prevent screening pigment granules of the retinular cells to reach the rhabdom edge and contribute to isolating an ommatidial cluster of retinular cells from neighboring ommatidial groups of retinular cells. In our samples, the narrow sleeve of black screening pigment granules was present directly around the edge of the rhabdom with an electron-empty palisade that creates a sleeve of lower refractive index further peripherally around the rhabdom. Such constellation would not necessarily improve the sensitivity of eyes under dim light [43], but would reduce optical overlap between neighboring ommatidia and thereby sharpen an image. Some unusual cellular organelles of vesicular nature were found in the vicinity of the palisade, which may be regarded as evidence of high metabolic activity in the retinular cells [43,48]. 

In addition to a pair of oval-shaped compound eyes, *A. similis* also possess triplets of dorsal ocelli. Some recent studies have shown that the ocellar photoreceptors of Hymenoptera are sensitive to polarized light [42,49], and can perceive ultraviolet light and green light [50,51]. These studies suggest that the ocelli of *A. similis* may play an important role in navigation. Behavioral experiments to cover the ocelli with opaque black paint could shed some light on the function of the ocelli, but this task is challenging because of the small size of the ocelli.

## Figures and Tables

**Figure 1 insects-13-00152-f001:**
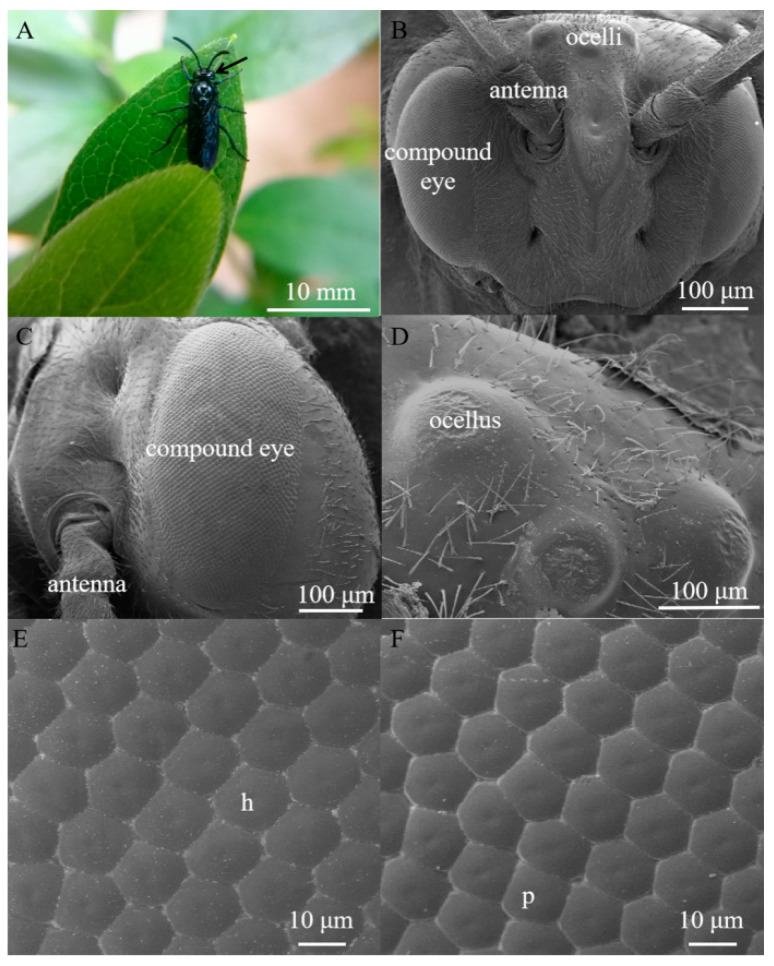
(**A**) Female adult of *Arge similis*, with compound eye position indicated by the black arrow. (**B**) Scanning electron micrograph of frontal view of the head, showing the position of compound eyes and ocelli. (**C**) Scanning electron micrograph of the lateral view of the compound eye showing little variation of sizes across the eye. (**D**) Scanning electron micrograph of the dorsal ocelli. (**E**) A close-up micrograph of the array of mostly hexagonal facets (h). (**F**) A close-up micrograph with some irregularly shaped and pentagonal facets (p).

**Figure 2 insects-13-00152-f002:**
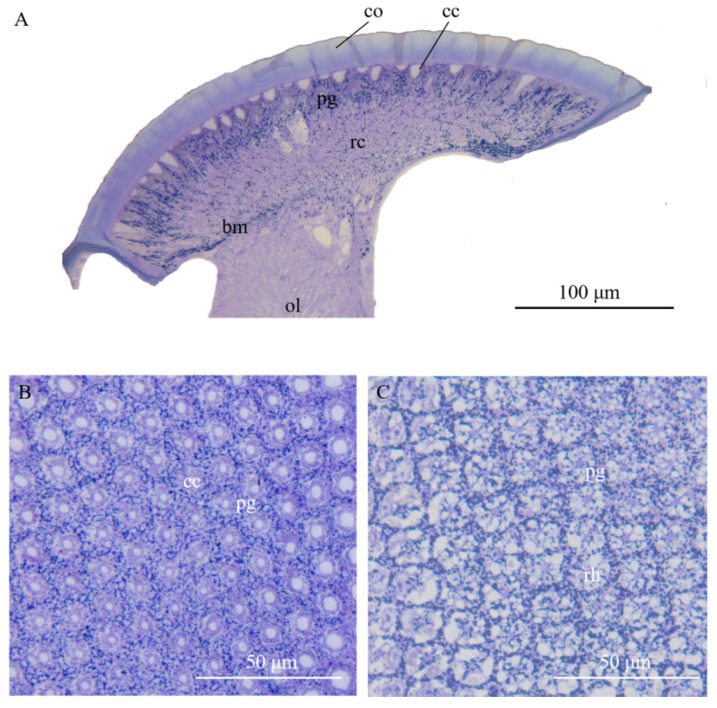
Light micrographs (LM) at different levels of the compound eye of *Arge similis*. (**A**) Longitudinal section of the compound eye in caudal-rostral direction: left to right. (**B**) Transverse section of the crystalline cones surrounded by pigment cells. (**C**) Transverse section of the retinular cell area, showing centrally fused rhabdoms and retina screening piments around the rhabdom edge and secondary pigment cell granules in peripheral positions between neighboring ommatidia. co: cornea; cc: crystalline cone; pg: pigment particles; rc: retinular cell; ol: optic lobe; rh: rhabdom; bm: basement membrane.

**Figure 3 insects-13-00152-f003:**
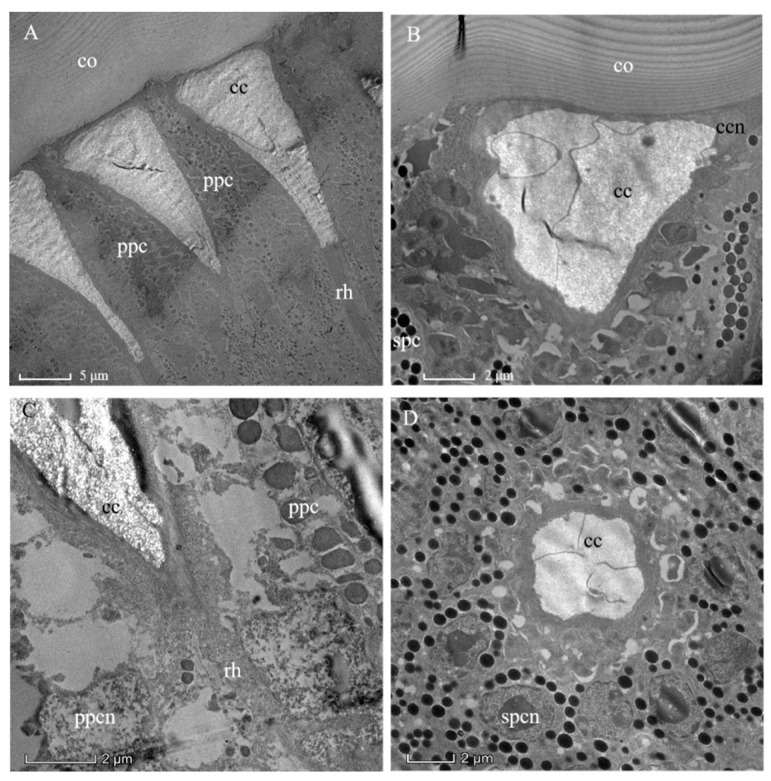
Transmission electron micrographs through the distal ommatidial region of the eye of a female *Arge similis*. (**A**) Part of the cornea and a longitudinally sectioned crystalline cone, rhabdoms, and retinular cell layer are shown. (**B**) Oblique section through cornea and crystalline cone transition zone, showing cone cell nuclei and primary, as well as secondary, pigment cells. (**C**) Longitudinal section of the cone-rhabdom contacting area, showing the proximal end of the cone enveloped by two primary pigment cells and the distal tip of the rhabdom. (**D**) Transverse section of the cone, the two primary pigment cells with severely discolored pigment grains and secondary pigment cells with spherical screening granules enveloping the central matrix of the crystalline cone. co: cornea; cc: crystalline cone; ccn: nuclei of cone cell; ppc: primary pigment cell; spc: secondary pigment cells; ppcn: nuclei of primary pigment cell; spcn: nuclei of second pigment cell; rh: rhabdom.

**Figure 4 insects-13-00152-f004:**
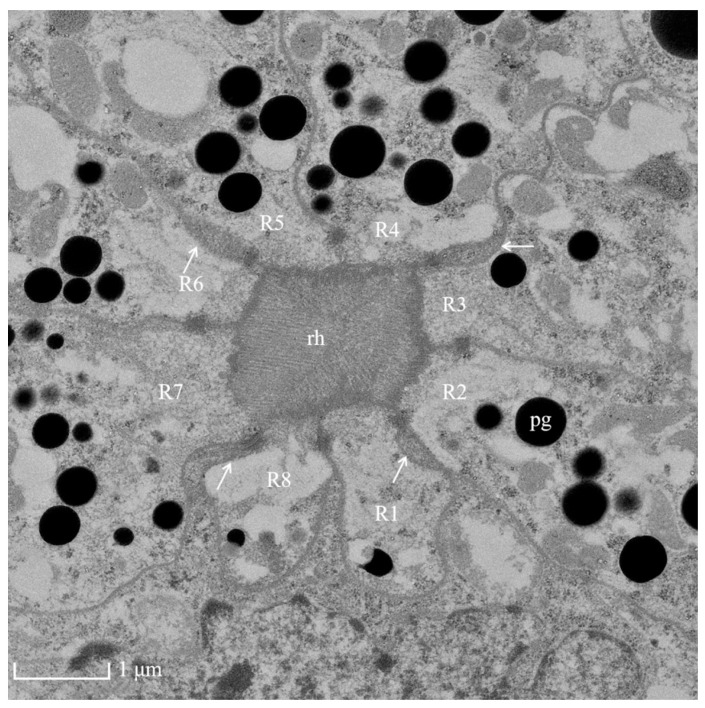
TEM micrograph of the central region of the rhabdom of the ommatidia of female *Arge similis* showing photoreceptive layer consisting of eight retinular cells (R1 to R8). Four cone roots are located between the retinular cell membranes (arrows). pg: pigment; rh: rhabdom.

**Figure 5 insects-13-00152-f005:**
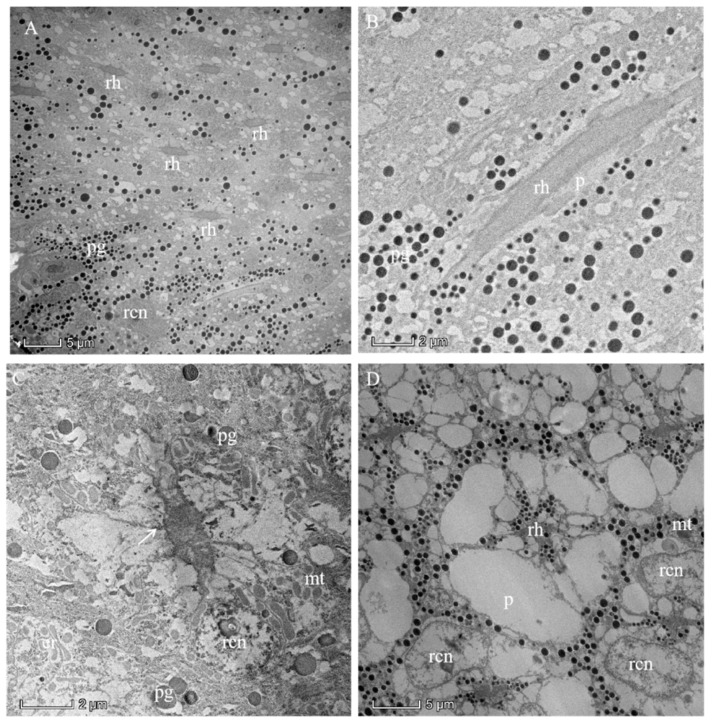
Proximal region of the ommatidial retina of the eye of a female *Arge similis* and the rhabdom in the dark-adapted state of adaptation. (**A**) Cross-section through different planes of the rhabdoms. (**B**) Longitudinal section of a rhabdom fragment surrounded by retinular cell cytoplasm containing small spherical pigment granules and secondary pigment cells with larger pigment grains. (**C**) Transverse section of centrally fused rhabdom, eight retinular cells containing a large number of mitochondria, and rhabdomeres that are separated by desmosomes (arrow) from neighboring rhabdomeres. (**D**) Transverse section of a rhabdom that is composed of retinular cells, whose rhabdomeres are flanked by featureless palisade. The pigment granules are densely clustered near the rhabdom. rh: rhabdom; pg: pigment; rcn: nuclei of retinular cell; mt: mitochondrion; er: endoplasmic reticulum; p: palisade.

**Figure 6 insects-13-00152-f006:**
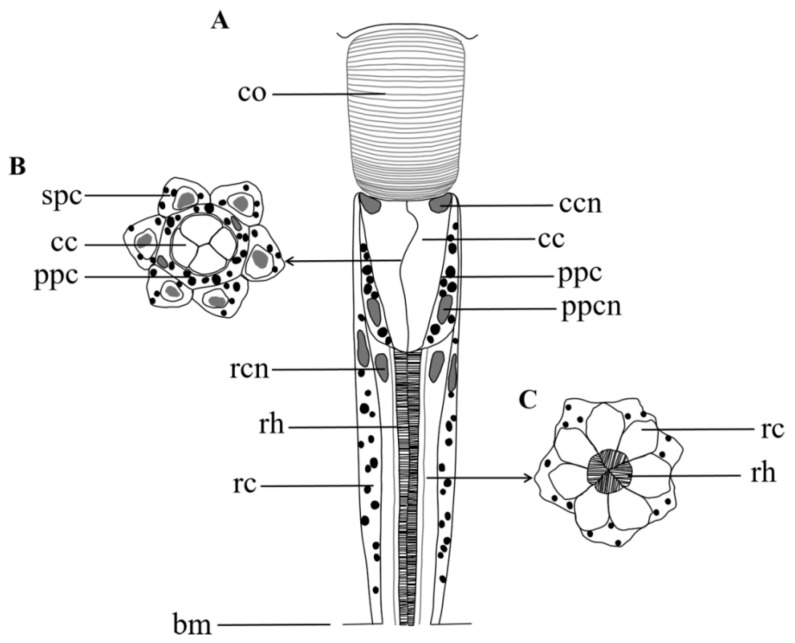
Semischematic drawing of one ommatidium of *Arge similis*. (**A**) Longitudinal view of ommatidium. (**B**) Transverse section through the region of the crystalline cone. (**C**) Transverse section through the region of the retinular cells and the rhabdom. co: cornea; cc: crystalline cone; ccn: nuclei of cone cell; ppc: primary pigment cell; spc: secondary pigment cells; ppcn: nuclei of primary pigment cell; rc: retinular cell; rcn: nuclei of retinular cell; rh: rhabdom; bm: basement membrane.

**Table 1 insects-13-00152-t001:** Morphological and histological details of the compound eyes in *Arge similis*. Number of samples (N), range, and averages (mean ± SE) are given in connection with each measurement.

Parameter	N	Unit	Range	Average
Facet number in males per eye	3	-	1819–2195	2022 ± 89
Facet number in females per eye	3	-	2095–2288	2223 ± 52
Compound eye major axis diameter in males	3	μm	1129.10–1225.70	1165.67 ± 24.70
Compound eye major axis diameter in females	3	μm	1173.58–1210.60	1192.20 ± 8.73
Compound eye minor axis diameter in males	3	μm	549.89–666.20	607.11 ± 27.43
Compound eye minor axis diameter in females	3	μm	595.29–611.37	603.66 ± 3.80
Compound eye area in males	3	mm^2^	0.58–0.60	0.59 ± 0.01
Compound eye area in females	3	mm^2^	0.55–0.65	0.60 ± 0.02
Facet diameter in males	120	μm	10.06–25.75	17.40 ± 0.19
Facet diameter in females	120	μm	13.32–22.86	18.26 ± 0.17
Crystalline cone length	3	μm	20.30–23.38	22.10 ± 0.76
Diameter of PPC pigment granules	50	μm	0.23–1.01	0.60 ± 0.02
Diameter of SPC pigment granules	50	μm	0.22–0.63	0.41 ± 0.01
Diameter of Ret C pigment granules	50	μm	0.15–0.68	0.34 ± 0.02
Diameters of median ocelli	3	μm	86.45–122.68	106.92 ± 8.75
Diameters of ocelli on both sides	3	μm	78.89–97.22	87.14 ± 4.39

## Data Availability

The raw data supporting the conclusions of this manuscript will be made available by the authors, without undue reservation, to any qualified researcher.

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
