# Peer review of "Fine Structure of the Visual System of Arge similis (Hymenoptera, Argidae)"

_insects, 2022, doi:10.3390/insects13020152_

Round 1

Reviewer 1 Report

I think this is a well written and interesting paper. I did have a few grammatical/ typo comments (noted in attached pdf), but once corrected the paper should be ready to publish. 

Besides: Please check that there are spaces before and after the reference brackets. The spaces appear to be missing in many references throughout the paper.

Author Response

Dear Editor and Reviewers, 

We sincerely thank you for valuable comments on our manuscript. We carefully went through your comments and revised the manuscript accordingly. We are resubmitting the revised manuscript. Following is our response to the comments:

Reviewer 1: I think this is a well written and interesting paper. I did have a few grammatical/ typo comments (noted in attached pdf), but once corrected the paper should be ready to publish. Besides: Please check that there are spaces before and after the reference brackets. The spaces appear to be missing in many references throughout the paper.

Response: We appreciate the valuable comments from the reviewer. We have carefully revised the MS based on the reviewers’ comments.

Reviewer 2 Report

The article is a detailed study of morphology of the compound eyes of the sawfly Arge similis  with light microscopy, scanning electron microscopy, and transmission electron microscopy. The results could potentially have a practical significance, since A. similis is a pest of Rhododendron plants. Photographs and analysis are of good quality. I would recommend making some minor corrections in the texts:

General comments:

  1. Please, indicate in the introduction if Arge similis is nocturnal or diurnal.
  2. Why compound eye in female has more ommatidia than in male? It would be interesting to discuss it in Discussion section. Do females find the place for oviposition with their eyes?
  3. It would also be interesting to discuss if similis mainly with eyes or also with other organs of sences.

Specific comments:

Line 17-18. Probably this sentence about the ommatidium structure could be replaced by an information about the idea of the article. The general structure of ommatidium is well known.

Line 24., Line 155 etc. If a Latin name is in the beginning of the sentence, it should be written in full. Please, replace “A. similis” with  “Arge similis”.

Line 40. (and all over the manuscript) Please, check the intervals before and after the brackets. They are missed in some places.

Line 57 Do adults also feed on Rhododendron leaves?

Line 79. How the gender was identified?

Line 80. Did you sever the heads from alive insects?

Line 125 Why is this formula in bold?

Line 131 Please, check in the author guidelines if the references to figures and table should be in bold or not.

Line 312 Please, correct: “would would”.

Author Response

Dear Editor and Reviewers, 

We sincerely thank you for valuable comments on our manuscript. We carefully went through your comments and revised the manuscript accordingly. We are resubmitting the revised manuscript. Following is our response to the comments:

Reviewer 2:

The article is a detailed study of morphology of the compound eyes of the sawfly Arge similis with light microscopy, scanning electron microscopy, and transmission electron microscopy. The results could potentially have a practical significance, since A. similis is a pest of Rhododendron plants. Photographs and analysis are of good quality. I would recommend making some minor corrections in the texts:

General comments:

  1. Please, indicate in the introduction if Arge similis is nocturnal or diurnal.

Response: Arge similis is diurnal. We mentioned this in the Introduction of the revised MS:

Arge similis is a diurnal insect (i.e., the adult is active during the day time) that feed on Rhododendron leaves to satisfy nutritional requirements for reproduction [17,21,22]”

  1. Why compound eye in female has more ommatidia than in male? It would be interesting to discuss it in Discussion section. Do females find the place for oviposition with their eyes?

Response: We discussed this point in the revised MS:

“Interestingly, the number of ommatidia and the area of compound eyes of females are larger than that of males in A. similis. Large compound eyes indicate that the insects can be expected to exhibit a high-quality vision ability [38,39,40], and may help gravid females to perceive suitable host plants to satisfy nutritional requirements and oviposition.” 

  1. It would also be interesting to discuss if similis mainly with eyes or also with other organs of sences.

Response: We discussed this point in the revised MS:

“In addition, some Lepidoptera insects search host plants through a combination of visual and olfactory stimuli [41]. A previous study showed that A. similis females had obvious electrophysiological responses to four compounds of volatiles of R. pulchrum, indicating that the olfactory organ may also play an important role in finding a host for A. similis females [21].” 

  1. Line 17-18. Probably this sentence about the ommatidium structure could be replaced by an information about the idea of the article. The general structure of ommatidium is well known.

Response: We rewrote the sentence as below:

“Each ommatidium consists of a lamellated cornea, a crystalline cone with eucone type and a centrally fused rhabdom composed of more than seven retinula cells.”

  1. Line 24., Line 155 etc. If a Latin name is in the beginning of the sentence, it should be written in full. Please, replace “ similis” with “Arge similis”.

Response: We changed “A. similis” to “Arge similis” in the beginning of the sentence throughout the manuscript.

  1. Line 40. (and all over the manuscript) Please, check the intervals before and after the brackets. They are missed in some places.

Response: We checked the intervals throughout the manuscript.

  1. Line 57 Do adults also feed on Rhododendron leaves?

Response: Adults also feed on Rhododendron leaves. We mentioned this in the Introduction of the revised MS:

A. similis is a diurnal insect (i.e., the adult is active during the day time) that feed on Rhododendron leaves to satisfy nutritional requirements for reproduction [17,22]”

  1. Line 79. How the gender was identified?

Response: We mentioned this in the Introduction of the revised MS:

“The gender of A. similis was differentiated using the method provided by Shi [2,19]. In brief, the end of the abdomen of female adults has a small gap, whereas the gap is absent for male adults [21].”

  1. Line 80. Did you sever the heads from alive insects?

Response: We sever the heads from alive insects. We mentioned this in the Introduction of the revised MS.

  1. Line 125 Why is this formula in bold?

Response: We changed the formula.

  1. Line 131 Please, check in the author guidelines if the references to figures and table should be in bold or not.

Response: We checked bold place throughout the manuscript.

  1. Line 312 Please, correct: “would would”.

Response: We deleted “would” in the revised manuscript.

Reviewer 3 Report

Please see attached

Author Response

Dear Editor and Reviewers, 

We sincerely thank you for valuable comments on our manuscript. We carefully went through your comments and revised the manuscript accordingly. We are resubmitting the revised manuscript. Following is our response to the comments:

Reviewer 3:

The manuscript describes the visual system of Arge similis. This is in principle interesting and would contribute to our knowledge of compound eye properties in Hymenoptera. However, at this stage, there are a number of issues with this manuscript which would need to be addressed in a major revision.

The required revision is major because it will probably require additional histological work which will need to pay more attention to the planes of sectioning. In particular, it is important to produce LM and TEM sections that are parallel or perpendicular to the long axis of ommatidia, for important measurements, such as the length and diameter of rhabdoms, for identifying the retinular cells and for assessing microvilli directions.

Response: We appreciate the valuable comments from the reviewer. We took multiple photos of LM and TEM sections and recalculated the measurements with the sections parallel or perpendicular to the long axis of ommatidia. We mentioned this in the revised manuscript.We also changed Fig 5C to show the microvilli directions:

The authors should also consider to use a more reliable method to estimate facet numbers.

Response: In our SEM test, the compound eyes were perpendicular to the SEM camera, and therefore the sample can be fully presented in the photos. We counted all ommatidiums on the picture. The photo used to measure the number of ommatidiums is shown as follows:We also provided a Figure S2 to show the measurements of the number of ommatidiums as below:

Figure S2. The measurements of the number of ommatidiums of Arge similis with scanning electron micrograph.

Major comments:

  1. Measurements of interommatidial angles and length of rhabdoms: Fig. 2A is a longitudinal section through the eye which is not parallel to the long axis of ommatidia and therefore cannot be used to establish interommatidial angles and length of rhabdoms. It is not clear how the length of crystalline cones and of rhabdoms was determined.

Response: We used eye radius to estimate interommatidial angles. Three LM sections of compound eye were used to determine the eye radius. We deleted “This is similar to the angle confirmed by direct measurements (4.90 ± 0.31°) from longitudinal sections of the eye (Figure 2A).” We measured length of crystalline cones and cornea again with the photos parallel to the axis.

  1. This is also true for estimating interommatidial angles from determining the local eye radius. The wasp’s eye is very elongated in the dorso-ventral direction and therefore must have a very different eye radius (and interommatidial angles) in horizontal and vertical directions. It is not clear which local eye radius was used to estimate interommatidial angles.

Response: We used LM sections in three different places of the compound eye to determine the distance from the top of the eye to the baseline (70.92, 80.542, 82.41μm) and baseline length of eye (321.187, 328.463, 336.506 μm). We mentioned this in the revised manuscript:

“It is worth noting that the eyes of A. similis is very elongated in the dorso-ventral direction, and their eye radius (and interommatidial angles) in horizontal and vertical directions are quite different. Therefore, we used LM sections in three different places of the compound eye to determine the distance from the top of the eye to the baseline and baseline length of the eye.”

  1. It is very difficult to count facet numbers in SEM images of a heavily curved compound eye. It would be better to use the nail-varnish technique (see Schwarz et al. 2011).

Response: In our SEM test, The compound eyes were perpendicular to the SEM camera, and the sample can be fully presented in the photos. In this study, we counted all ommatidiums on the picture.

  1. TEM sections in Fig. 3A-C and Fig. 4B are not parallel to the long axis of the ommatidia. The same problem exists for those TEM sections that should be perpendicular to the long axis of ommatidia (e.g. Fig. 4A and Fig. 5A).

Response: We agree with the reviewer. These figures can cannot be used for measurements. We deleted the measurement data using these pictures, and we recalculated the measurements on the figures that parallel or perpendicular to the long axis of the ommatidia.

  1. Fig. 5B & C are not of sufficient quality to allow the reader to distinguish cell boundaries, to verify the labelling of retinular cells and to verify the statements about microvilli directions made in the text.

Response: We changed Fig. 5B and Fig. 5C.

  1. Check inappropriate statements/figure references in the text that are not supported by what figures show:

Response: We carefully checked the the statements in the text.

  1. Line 200: Fig. 3C does not show the full length of the crystalline cone.

Response: We changed “Fig. 3C” to “Fig. 3B.” 

  1. Line 228: Fig. 5C does not show length of rhabdoms.

Response: We deleted “to create a rhabdom with a length of 35.69 ± 1.77 μm”.

  1. Line 230: Fig. 5C does not show that the cisternae ‘surrounds the rhabdom over mostof its length’.

Response: We deleted “surrounds the rhabdom over most of its length”.

  1. Line 230: There is no Fig. 5D to show screening pigment movements.

Response: We changed “Fig. 5D” to “Fig. 5B.” 

  1. Line 261: Fig. 3B does not support this statement.

Response: We deleted “The SPCs reach from the cornea down to the basement membrane and fill the spaces between adjacent ommatidia (Figure 3B)”.

  1. Line 309: This statement cannot be verified in the supplied images.

Response: In the figure 4 and figure 5, we found considerable palisade around the rhabdom.

  1. Minor comments:

Change retina/retinula cells to retinular cells throughout.

Response: We changed “retina/retinula cells” to “retinular cells” throughout the manuscript.

Round 2

Reviewer 3 Report

see attached comments in red.

Author Response

ear Editor and Reviewers, 

We sincerely thank you for valuable comments on our manuscript. We carefully went through your comments and revised the manuscript accordingly. We are resubmitting the revised manuscript. Following is our response to the comments:

Reviewer 3:

The manuscript describes the visual system of Arge similis. This is in principle interesting and would contribute to our knowledge of compound eye properties in Hymenoptera. However, at this stage, there are a number of issues with this manuscript which would need to be addressed in a major revision.

The required revision is major because it will probably require additional histological work which will need to pay more attention to the planes of sectioning. In particular, it is important to produce LM and TEM sections that are parallel or perpendicular to the long axis of ommatidia, for important measurements, such as the length and diameter of rhabdoms, for identifying the retinular cells and for assessing microvilli directions.

Response: We appreciate the valuable comments from the reviewer. We took multiple photos of LM and TEM sections and recalculated the measurements with the sections parallel or perpendicular to the long axis of ommatidia. We mentioned this in the revised manuscript. We also changed Fig 5C to show the microvilli directions:

You should show the sections from which you took measurements, in particular of

length of crystalline cone and length of rhabdom.

Response: We appreciate the valuable comments from the reviewer. We changed the Fig 3A to show the length of crystalline cone. As suggested by the reviewer, we currently do not have a picture to clearly show the length of rhabdom, therefore we deleted the measurement of length of rhabdom in the manuscript.

This is a low-quality and skew section that does not allow one to see cell boundaries

of retinular cells. It is also not clear at what level of the ommatidium this section was

Taken.

Response: We agree with the reviewer that this section is not quite clear. We deleted this section and the description of this section in the manuscript.

The authors should also consider to use a more reliable method to estimate facet numbers.

Response: In our SEM test, the compound eyes were perpendicular to the SEM camera, and therefore the sample can be fully presented in the photos. We counted all ommatidiums on the picture. The photo used to measure the number of ommatidiums is shown as follows:We also provided a Figure S2 to show the measurements of the number of ommatidiums as below:

Figure S2. The measurements of the number of ommatidiums of Arge similis with scanning electron micrograph.

Thank you. I am confident about your facet number measurements now

Major comments:

  1. Measurements of interommatidial angles and length of rhabdoms: Fig. 2A is a longitudinal section through the eye which is not parallel to the long axis of ommatidia and therefore cannot be used to establish interommatidial angles and length of rhabdoms. It is not clear how the length of crystalline cones and of rhabdoms was determined.

Response: We used eye radius to estimate interommatidial angles. Three LM sections of compound eye were used to determine the eye radius. We deleted “This is similar to the angle confirmed by direct measurements (4.90 ± 0.31°) from longitudinal sections of the eye (Figure 2A).” We measured the length of crystalline cones and cornea again with the sections parallel to the axis.

  1. This is also true for estimating interommatidial angles from determining the local eye radius. The wasp’s eye is very elongated in the dorso-ventral direction and therefore must have a very different eye radius (and interommatidial angles) in horizontal and vertical directions. It is not clear which local eye radius was used to estimate interommatidial angles.

Response: We used LM sections in three different places of the compound eye to determine the distance from the top of the eye to the baseline (70.92, 80.542, 82.41μm) and baseline length of eye (321.187, 328.463, 336.506 μm). We mentioned this in the revised manuscript:

“It is worth noting that the eyes of A. similis is very elongated in the dorso-ventral direction, and their eye radius (and interommatidial angles) in horizontal and vertical directions are quite different. Therefore, we used LM sections in three different places of the compound eye to determine the distance from the top of the eye to the baseline and baseline length of the eye.”

You should show these sections. Fig. 2A is NOT parallel to the long axes of ommatidia.

Response: We tried to calculate the distance from the top of the eye to the baseline using three LM sections in three different places of the compound eye (show as below). Unfortunately, as suggested by the reviewer, these three photos are not completely parallel or perpendicular to long axis of ommatidia. As a result, the interommatidial angles calculated based on these measurements might be bias. We decided to remove these parameters from the manuscript. We sincerely thank the reviewer for pointing out mistakes in our manuscript.

Figure S3 Three LM sections in three different places of the compound eye

  1. It is very difficult to count facet numbers in SEM images of a heavily curved compound eye. It would be better to use the nail-varnish technique (see Schwarz et al. 2011).

Response: In our SEM test, The compound eyes were perpendicular to the SEM camera, and the sample can be fully presented in the photos. In this study, we counted all ommatidiums on the picture.

Thank you.

  1. TEM sections in Fig. 3A-C and Fig. 4B are not parallel to the long axis of the ommatidia. The same problem exists for those TEM sections that should be perpendicular to the long axis of ommatidia (e.g. Fig. 4A and Fig. 5A).

Response: We agree with the reviewer. These figures can cannot be used for measurements. We deleted the measurement data using these pictures, and we recalculated the measurements on the figures that parallel or perpendicular to the long axis of the ommatidia.

  1. Fig. 5B & C are not of sufficient quality to allow the reader to distinguish cell boundaries, to verify the labelling of retinular cells and to verify the statements about microvilli directions made in the text.

Response: We changed Fig. 5B and Fig. 5C.

  1. Check inappropriate statements/figure references in the text that are not supported by what figures show:

Response: We carefully checked the the statements in the text.

  1. Line 200: Fig. 3C does not show the full length of the crystalline cone.

Response: We changed “Fig. 3C” to “Fig. 3B.” 

Fig. 3B also does not show the full length of the crystalline cone.

Response: We provided Fig3A to show the full length of the crystalline cone.

  1. Line 228: Fig. 5C does not show length of rhabdoms.

Response: We deleted “to create a rhabdom with a length of 35.69 ± 1.77 μm”.

  1. Line 230: Fig. 5C does not show that the cisternae ‘surrounds the rhabdom over mostof its length’.

Response: We deleted “surrounds the rhabdom over most of its length”.

  1. Line 230: There is no Fig. 5D to show screening pigment movements.

Response: We changed “Fig. 5D” to “Fig. 5B.” 

That figure also does NOT show pigment movement depending on adaptation.

Response: We agree with the reviewer that the figure does not show pigment movement. We rewrote the sentence as below:

“The distal pigment granules were far away from the rhabdom under normal light intensity (Figure 4C), while screening pigment granules of retinula cell were near the rhabdom during dark adaptation (Figure 4D).”

“Fig. 5D” to “Fig. 5B.”

  1. Line 261: Fig. 3B does not support this statement.

Response: We deleted “The SPCs reach from the cornea down to the basement membrane and fill the spaces between adjacent ommatidia (Figure 3B)”.

  1. Line 309: This statement cannot be verified in the supplied images.

Response: In the figure 4 and figure 5, we found considerable palisade around the rhabdom.

  1. Minor comments:

Change retina/retinula cells to retinular cells throughout.

Response: We changed “retina/retinula cells” to “retinular cells” throughout the manuscript.

Round 3

Reviewer 3 Report

Ommatidial dimensions and the rhabdomere contributions to the rhabdom are important aspects in describing a new compound eye. You will need to present carefully prepared LM and TEM sections that are truly parallel and perpendicular to the long axes of ommatidia. Your new Fig. 3A is good, but for the rest of the manuscript you will need to do more histological work. After all, three months worth of sectioning and measurements are a worth while exercise to produce a high quality paper, which is not unusual for a major revision.

Author Response

Dear Editor and Reviewers, 

We sincerely thank you for valuable comments on our manuscript. We carefully went through your comments and revised the manuscript accordingly. We are resubmitting the revised manuscript. Following is our response to the comments:

Reviewer 3

Ommatidial dimensions and the rhabdomere contributions to the rhabdom are important aspects in describing a new compound eye. You will need to present carefully prepared LM and TEM sections that are truly parallel and perpendicular to the long axes of ommatidia. Your new Fig. 3A is good, but for the rest of the manuscript you will need to do more histological work. After all, three months worth of sectioning and measurements are a worth while exercise to produce a high quality paper, which is not unusual for a major revision.

Response: We revised the Fig. 2A and remeasured the ommatidial dimensions based on the sections that are perpendicular to the long axes of ommatidia.

Fig. 2A:  Section perpendicular to the long axes of ommatidia.

We also revised Fig. 5 to show the rhabdomere contributions .

Figure 5. TEM micrograph of the central region of the rhabdom of the ommatidia of female Arge similis.

the authors present important measurements regarding the functional anatomy of this wasp compound eye, but the images they provide do not allow one to make these measurements. I therefore asked the authors to provide the following:
Horizontal and vertical light microscopy sections through the compound eye that are parallel to the long axes of ommatidia. Such sections would allow one to verify their measurements of the length of crystalline cones and the length of rhabdoms.
The example sections they had provided so far in Fig. 2A were not parallel to the long axes of ommatidia, showed signs of tissue distortion and tearing and the section orientation (horizontal/vertical) was not clear.

Response: We deeply appreciate your valuable comments that significantly improve our MS. We tried our best to prepare the semithin sections. However, we found it is difficult to obtain the sections through the compound eye that are completely parallel to the long axes of ommatidia. For the round-shaped compound eyes, multiple sections through the long axis can be cut (Fig 1). However, the compound eyes of Arge similis are oval-shaped. Therefore, there is only one section through the long axis of the eye (Fig 2). We cut many sections, and even very small deviations of the angle may make the sections unparallel to the long axes of ommatidia. Unfortunately, it is not possible to verify whether the section is completely vertical and through the central axis long axis until they are stained and observed under a microscope. Therefore, no cues are available for fine adjustment during section cutting. We have cut all compound eye samples stored in our lab, but no section can reach the requirement of the reviewer.
We agree with the reviewer that the currently available sections cannot allow us to correctly measure the length of crystalline cones and rhabdoms. As suggested by reviewer 3, we removed such measurements and relevant statements. However, we still think the remaining results could contribute to understand the structure and characteristics of the vision system of A. similis.

Round 4

Reviewer 3 Report

see attached

Author Response

We sincerely thank you for valuable comments on our manuscript.  Following is our response to the comments:

Thank you for considering my concerns and suggestions. From your Fig. 2, I can see now what your problem is: the compound eye of A. similis has a very unusual facet arrangement. In large parts of the eye, the facets are not arranged in a hexagonal pattern. It is worth pointing this out. Why not include the Fig. 2 from your letter in the paper?

I agree that it would be difficult to find a section plane that is aligned with a row of facets (which would be important for anatomical estimates of interommatidial angles). This said, however, it would still have been possible to cut sections in vertical (as indicated) and horizontal directions (relative to head coordinates) that would have been parallel to the long axes of some of the ommatidia, if not neighbouring ones (to measure crystalline cone lengths and the lengths of rhabdoms).

If you do not want to take the time or do not have the opportunity to prepare better sections, I will not stand in the way of you publishing what you have at this stage. However, what does worry me is the fact that you initially presented measurements which you withdrew after I queried how you could make them. And you also made statements, such as pigment movement during light-dark adaptation, for which you did not show evidence.

Response:
Question 1: Why not include the Fig. 2 from your letter in the paper?
Response: The Fig. 2 from the letter is Fig. 1C in the paper.
Question 2: And you made statements, such as pigment movement during light-dark adaptation, for which you did not show evidence?
Response: In the paper, we wrote “the screening pigment granules of retinula cell were near the rhabdom during dark adaptation.” but not pigment movement. The Fig 4D can show the pigment near the rhabdom.
